# Analysis of Spatiotemporal Characteristics and Influencing Factors of Electric Vehicle Charging Based on Multisource Data

Chenxi Liu [1,2], Zhenghong Peng [1,2], Lingbo Liu [1,2,3] and Hao Wu [1,2,*]

1 Department of Urban Planning, School of Urban Design, Wuhan University, Wuhan 430072, China; yololiuchenxi@whu.edu.cn (C.L.); pengzhenghong@whu.edu.cn (Z.P.); lingbo.liu@whu.edu.cn (L.L.)
2 Digital City Research Center, School of Urban Design, Wuhan University, Wuhan 430072, China
3 Center for Geographic Analysis, Harvard University, Cambridge, MA 02138, USA
* Correspondence: wh79@whu.edu.cn

**Abstract:** Amid the global shift towards sustainable development, this study addresses the burgeoning electric vehicle (EV) market and its infrastructure challenges, particularly the lag in public charging facility development. Focusing on Wuhan, it utilizes big data to analyze EV charging behavior's spatiotemporal aspects and the urban environment's influence on charging efficiency. Employing a random forest regression and multiscale geographically weighted regression (MGWR), the research elucidates the nonlinear interaction between urban infrastructure and charging station usage. Key findings include (1) a direct correlation between EV charging patterns and urban temporal factors, with notable price elasticity; (2) the predominant influence of commuting distance, supplemented by the availability of fast-charging options; and (3) a strategic proposal for increasing slow-charging facilities at key urban locations to balance operational costs and user demand. The study combines spatial analysis and charging behavior to recommend enhancements in public EV charging infrastructure layouts.

**Keywords:** public charging stations; spatiotemporal characteristics of charging behavior; random forest; SHAP; MGWR; optimization of charging station layout

## 1. Introduction

China committed to achieving carbon neutrality by 2060 during the 75th United Nations General Assembly in 2020 [1]. The transportation sector, being the third-largest energy consumer in China [2], also stands as a major source of carbon emissions. Electric vehicles, as an emerging eco-friendly mode of transportation, offer the potential to significantly reduce carbon emissions and urban air pollution, thereby optimizing urban energy structures and fostering sustainability [3]. Internationally, numerous countries have recognized the electrification of transportation as a pivotal strategy for mitigating greenhouse gas emissions. In a recent report published by the International Energy Agency, it was revealed that, in 2020, the worldwide new electric vehicle registration count reached a record-breaking 3 million, marking a 41% increase from the previous year. Notably, the cumulative new electric vehicle registrations in Europe and China accounted for over 80%. Furthermore, in the first quarter of 2021, global electric vehicle sales experienced a remarkable year-on-year increase of approximately 140% [4]. Among them, the number of new energy vehicles in China has reached 18.21 million, accounting for 5.5% of the total number of vehicles. In 2023, the registration of new energy vehicles accounted for 28.6% of new car registrations. On the other hand, the sharp rise in the price of oil use and the low cost of charging have led many drivers with average income to prefer the use of electric vehicles, which further promotes the prosperity of the electric vehicle market. With the rapid expansion of the electric vehicle market, a series of usage barriers, including limited driving range, inadequate charging infrastructure, and extended charging durations, have become increasingly conspicuous [5]. This trend necessitates fresh considerations for the

planning of the public charging station (PCS) network. The optimization of public charging facility construction can effectively alleviate range anxiety among electric vehicle drivers, a factor of utmost significance for the future development of the electric vehicle market [6]. On the other hand, compared with fuel vehicles, electric vehicles have longer charging times and lower charging station coverage, making range anxiety and time trade-offs more pronounced for electric vehicle users. Therefore, in the development of charging stations, a comprehensive evaluation of user charging behavior characteristics is indispensable to ensure that these stations are aligned with user requirements and enhance their utilization efficiency, thus avoiding the wasteful allocation of resources.

Previous research has employed diverse data sources for investigating charging behavior and assessing potential demands, including GPS trajectory data, surveys, and vehicle registration data [7–9]. However, these data can only indirectly estimate charging demands, and errors may arise during the evaluation process due to variations in model parameters or conversion methods. In reality, the normal operation of charging stations in China necessitates unified access to the power grid system. The evolution of information and communication technology has facilitated the acquisition of more intuitive charging data through web mining, substantially enhancing the accuracy and scientific rigor of the analysis. Presently, within studies utilizing charging station data, the scale of data can only encompass a limited number of charging stations or a modest number of charging events [3,10], resulting in relatively low data granularity that may not fully capture charging behavior. In this study, we procured data from nearly 1000 charging stations via web mining, incorporating a substantial volume of charging event data in the millions. Moreover, concerning data content, we ensured a sufficient level of detail, encompassing various charging pile usage information within each charging station, enabling a more intuitive and comprehensive exploration of user charging behavior.

Furthermore, research has demonstrated that the efficiency of public charging stations can be influenced by numerous factors, such as population density, road characteristics, location safety, and charging station operational conditions [11,12]. Consequently, in the planning and construction of charging stations, it is imperative not only to consider actual charging demands but also to account for the impact of the surrounding built environment on user charging preferences [13,14]. Presently, research concerning the influence of the built environment on charging station selection often lacks comprehensive consideration, and the selection of influencing factors may lack precision. In reality, charging behavior is shaped by three dimensions: the configuration of the charging stations themselves, the surrounding traffic conditions, and charging demands [13–15]. Traffic conditions encompass not only fundamental road characteristics but also spatial disparities in the placement of charging stations due to variations in road network structure. Charging demands include not only actual demands arising from the number of electric vehicles but also variations in charging frequency and pricing demands based on the intensity of usage needs (such as commuting) and economic capabilities. Consequently, it is imperative to comprehensively and scrupulously consider urban built environment indicators that may impact charging station utilization.

In the current context of investigating the influence of the built environment on the efficiency of charging station utilization, the predominant methods include the application of analytic hierarchy process (AHP) [11] and GIS-based regression models [16,17]. Nevertheless, due to the intricate network characteristics of urban PCS networks, this process, where the built environment affects charging choices, may entail nonlinear or sublinear relationships, along with threshold effects. These threshold effects manifest when an impact on usage efficiency becomes evident only once a factor surpasses a certain threshold. Consequently, relying solely on linear regression models proves inadequate. Machine learning, however, provides a viable solution for addressing these nonlinear phenomena. Tree-based algorithms such as gradient-boosted regression trees (GBRT), eXtreme gradient boosting (XGBoost), and random forest regression (RFR), in contrast to neural networks that solely predict outcomes, offer the capability to elucidate the intricate relationships

between influencing factors and the dependent variable [18,19]. On the other hand, machine learning necessitates a substantial number of parameters to achieve a high degree of accuracy, surpassing the requirements of traditional statistical models, which often suffice with only a few parameters. The output of machine learning models tends to be intricate and challenging to interpret. To interpret the roles of feature factors in the model, a method apart from Friedman's proposed visualization through partial dependence plots (PDP) [20] has been developed successfully through collaborative efforts, known as the SHAP (SHapley Additive exPlanations) algorithm [21,22]. Presently, SHAP is primarily employed in tandem with XGBoost ensembles [23], with fewer instances of its application in conjunction with other machine learning models. Furthermore, there exists a limited body of research that delves into the spatial interaction among factors and their impact on charging station usage. To surmount the constraints posed by traditional global regression models, stemming from their failure to account for spatial variables, and to address estimation biases arising from the spatial heterogeneity of different influencing factors, this study opts for multiscale geographically weighted regression (MGWR). This choice allows for the description of spatial heterogeneity in the influence coefficients of key factors within the study area and subsequently informs the formulation of optimization strategies.

This research leverages an extensive dataset of one million charging events to extract and analyze the spatiotemporal aspects of user charging behavior. Additionally, it introduces a factor system, based on multiple data sources, designed to evaluate the influence of the urban environment on the efficiency of charging stations. We employ RFR to investigate the nonlinear relationship between these variables, further complemented by the use of SHAP and PDP techniques to elucidate the inner workings of machine learning. Moreover, by examining the spatial interactions among crucial factors in the MGWR results, we propose optimization strategies for charging station layout. For urban planners and charging station operators, it is imperative to assess user charging behavior preferences and comprehend the correlation between the urban environment and charging station efficiency. Such insights are vital for enhancing PCS configuration and network layout, ultimately fostering the development of new energy vehicles and curbing urban carbon emissions. Our study contributes to this field in three significant ways. Firstly, it enhances the charging station database to a million-unit scale, offering a more precise description of user charging behavior. Secondly, it introduces a comprehensive and detailed urban environment indicator system to scrutinize its impact on charging station efficiency. Lastly, it provides a thorough exploration of the nonlinear relationship between these aspects and illustrates the application of XAI in charging station efficiency research. We also propose optimization strategies based on the spatial interactions of essential factors.

The structure of this paper is organized as follows: Section 2 introduces the study area and the utilized dataset, Section 3 outlines the applied methodologies, and Sections 4–6 present the results, analysis, and conclusions, respectively.

## 2. Study Area and Dataset

In this section, we will explain the reasons for choosing Wuhan as the research area. We will also provide a detailed explanation of the methods of obtaining research data, its specific content, and the processing procedures.

### 2.1. Study Area

We have chosen Wuhan as our research area to investigate the spatiotemporal patterns of user charging behavior and analyze the influence of the urban environment on the efficiency of charging stations. Wuhan, the largest central city in central China, was among the first cities to initiate pilot projects for new energy vehicles and possesses a robust foundation in the new energy industry. As of 2022, Wuhan has established a network of more than 1000 charging stations, and the average spacing between charging stations in the central urban area has been reduced to less than one kilometer. Hence, Wuhan offers a

highly suitable context for studying charging station usage. To ensure the precision of our research, we confined our investigation to the primary urbanized zone of Wuhan (Figure 1).

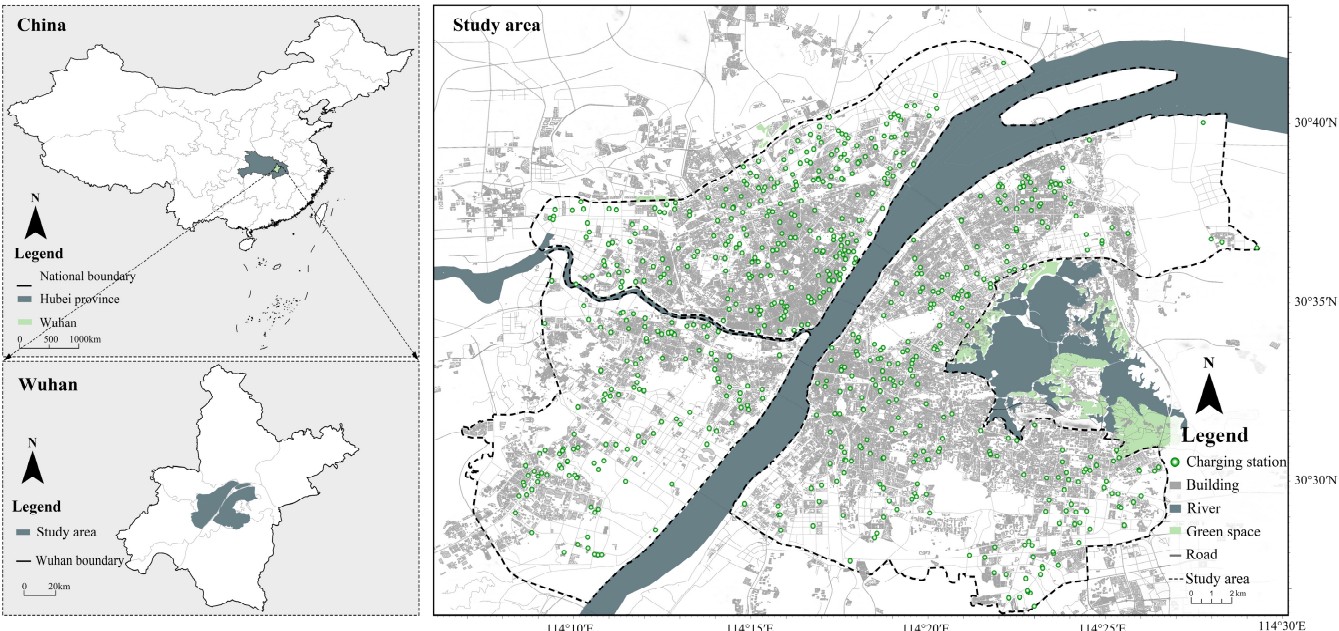

**Figure 1.** The location of Wuhan and the study area. Image source: Author's drawings.

### 2.2. *Data Source and Processing*

This study explores the use of charging stations based on multi-source data, which is conducive to enhancing the scientific nature of the research. When processing data, attention must be paid to data cleaning and the uniformity of the research scale.

### 2.2.1. Electric Vehicle Charging Data

The charging station usage data for this study were acquired using web scraping on the Python platform. The initial dataset comprises one million charging events from a total of 1029 charging stations. The data cover a three-month period, from 19 June 2023, to 17 September 2023. Due to the limitations of network mining, this study is unable to obtain information on all operational charging stations in Wuhan. The charging event dataset includes essential information, such as the charging station's unique ID and name, timestamp of data collection, pricing details, the total count of DC (direct current) and AC (alternating current) charging units, and the real-time utilization of both DC and AC charging units, as depicted in Table 1. Low-quality data can significantly compromise the accuracy of results, necessitating data cleaning to enhance data quality. Charging station data with zero counts for both DC and AC charging units are considered invalid. This condition signifies that the charging station did not interface for usage, rendering it irrelevant to our research; consequently, these data points have been excluded. Moreover, our study rigorously focused on publicly accessible charging stations while excluding data with nonzero values for "private pile sharing". Consequently, we identified 858 valid charging stations and 940,000 records of charging events. We performed geocoding on the names of the charging stations to obtain geographical spatial information.

**Table 1.** Charging station information description.

| Data Field | Description | Sampling Data |
|---|---|---|
| ID | Unique ID of the station | 1,485,167 |
| Name | Name of the charging station | "Wuhan University Doctoral Apartment Charging Station" |
| Capture time | Data scraping timestamp | 16 September 2023, 22:46:20 |
| Price | Real-time charging price | 0.6 |
| In-use DC chargers | | 1 |
| Total DC chargers | | 10 |
| In-use AC chargers | | 3 |
| Total AC chargers | | 8 |
| Private chargers shared | If the data are not zero, they represent private charging piles | 0 |

### 2.2.2. Mobile Signal Data

This study employs data obtained from a telecommunications company in Wuhan, China, in March 2017. The dataset includes user call records, which capture user calling behavior and information about base stations. To begin, we categorized travel data, extracted user age information, and conducted a summary analysis of the demographic age distribution within specific spatial units. Recognizing the high density of base stations in the central urban area, we filtered out data points with travel distances of less than 100 m. This process resulted in a dataset of 3,037,616 valid records and encompassed 1515 spatial units with valid data. Subsequently, we determined the spatial locations of user call stations and the corresponding time periods of user calls. Building upon this information, we demarcated spatial units representing user residences and workplaces, enabling us to summarize the population density within these temporal and spatial segments. Lastly, we computed the distances between these home and workplace spatial units, ultimately deriving the average commuting distance for the population within these designated spatial units. Detailed sample data are presented in Table 2.

**Table 2.** Sample mobile signal data.

| User ID | Base Station ID | Area ID | Travel Distance (m) |
|---|---|---|---|
| 2705381166310 | 2893326775 | A01010302 | 70,550.81 |
| 2705180683220 | 2872345954 | A01010502 | 71,476.55 |
| 2705277594080 | 2869955752 | A02010103 | 58,374.75 |

### 2.2.3. Urban Statistical Information and Geographic Data

Urban statistical data encompass housing price and consumer purchasing power information, while geographic data include point of interest (POI) data, application programming interface (API) travel records, and city road data. The original dataset is consolidated and summarized at the level of traffic analysis zones (TAZ), resulting in an attribute table that contains geographical spatial details. Housing price data are acquired through web scraping from designated websites using multiple IP addresses, while POI data are sourced from the Amap platform.

## 3. Methodology

In this section, we will focus on explaining the main model methods for constructing and applying the index system of influencing factors, according to the research approach.

### 3.1. Research Framework

Figure 2 depicts the workflow of this comprehensive study. (1) Multiple data sources are procured through methods such as web scraping. Employing TAZ as analytical units, separate databases are established for Wuhan's charging stations and the built environment factor system. (2) The spatiotemporal attributes of user charging behavior are diligently extracted and examined. (3) Appropriate machine learning models are selected to assess the spatial, nonlinear effects of the built environment on charging station efficiency. A spatial model is used to investigate the spatial heterogeneity of the most influential factors, and optimization ideas are proposed accordingly.

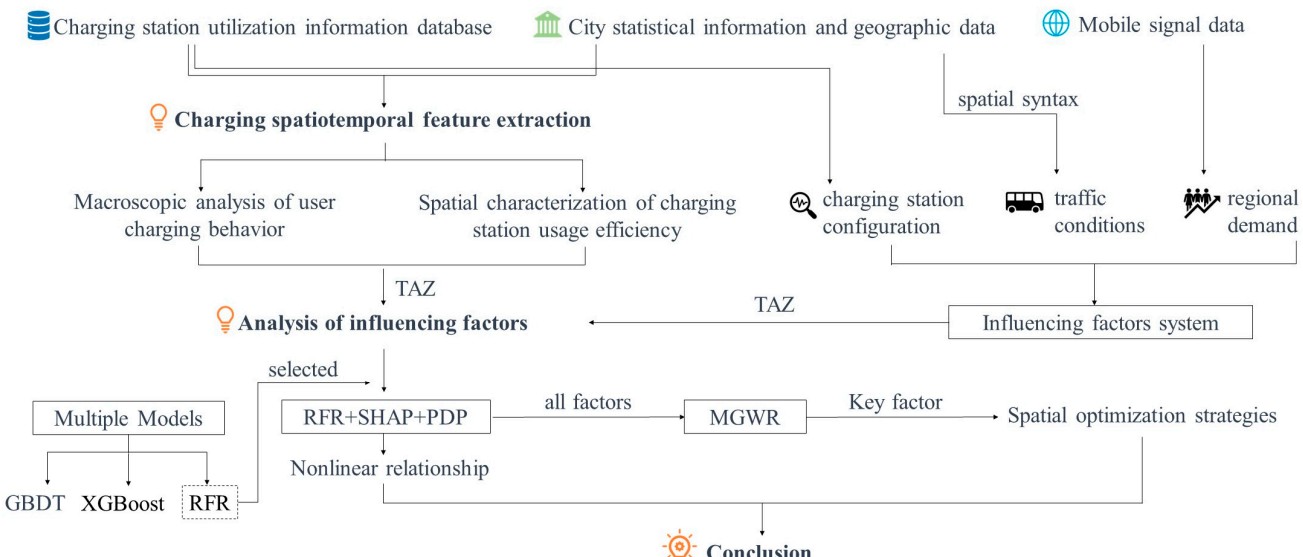

**Figure 2.** Research framework.

### 3.2. TAZ Generation and Influencing Factors System Construction

TAZ are vital for scrutinizing urban functional structures. In urban and transportation research, city blocks are commonly embraced as the quintessential analytical units [24]. Consequently, adhering to Wuhan's meticulous urban planning, traffic zones, totaling 2383, are designated as the fundamental spatial units for an in-depth examination of the central urban area.

Taking into full account the elements in the built environment that pertain to charging station utilization and drawing from extant literature [11–15], this paper formulates 12 indicators across three dimensions: charging station self-configuration, surrounding traffic conditions, and regional demand levels, with the objective of scrutinizing the impact of the built environment on charging station efficiency. The configuration of the charging station itself and its surrounding environment influence whether users choose to charge at that location, resulting in variations in the efficiency of each charging station. At the same time, the level of charging demand is an important aspect affecting the efficiency of charging station use. Therefore, we explore three major aspects: the supply side, the demand side, and the usage environment. Table 3 presents descriptions and calculation methodologies for all these indicators. All computed information is appended to the geographical attribute table of TAZ. Following this, a spatial linkage is established with the charging station database, retaining TAZ data with charging station distribution and discarding the remainder. This meticulous process culminates in the creation of the final research database.

**Table 3.** Impact factors and descriptions.

| Indicator | Description |
|---|---|
| charging station self-configuration | |
| X1: Charging Price | The average charging price for all charging piles at each station. |
| X2: DC Ratio | The ratio of DC chargers to all chargers. |
| X3: Operating Scale | The total number of charging piles at each charging station. |
| surrounding traffic conditions | |
| X4: Transportation attraction | The ability of a space to attract traffic as a destination, reflecting the centrality of the space in the entire system. |
| X5: Utilization potential | The advantages of a spatial unit as the shortest path for travel, reflecting the likelihood of a space being traversed. |
| X6: Metro accessibility | Obtained using Amap API for walking time from the centroids of various spatial units to |
| X7: Bus accessibility | the nearest subway or bus station. |
| regional demand levels | |
| X8: Population | Calculated using mobile signaling data to estimate potential usage intensity distribution. |
| X9: House price | Calculated by averaging housing prices within spatial units. |
| X10: Per capita consumption | Calculated using per capita restaurant consumption within spatial units. |
| X11: Commuting distance | Calculated using mobile signaling data. |
| X12: Functional mixing degree | Calculated by the entropy of POI categories within spatial units. |

### 3.3. Research Model and Main Algorithms

We utilize partial dependence plots (PDP) and SHAP (SHapley Additive exPlanations) to explain our predictive model, opening up the black box of machine learning. Moreover, we use multiscale geographically weighted regression (MGWR) to conduct spatial analysis on the most important features and propose relevant optimization strategies.

### 3.3.1. Random Forest Regression Model

The random forest model represents a machine-learning algorithm grounded in classification trees [25]. It is an ensemble learning-based algorithm widely used for regression problems. It models data using multiple decision trees and integrates their prediction results, thereby enhancing the model's performance and stability. This model exhibits notable algorithmic advantages, manifesting robustness against both noise and outliers. Its versatility extends to applications in clustering, classification, regression, and survival analysis, all the while facilitating the assessment of variable significance. Within the scope of this study, RFR was executed within the R language environment utilizing the "RandomForest" package.

### 3.3.2. Partial Dependency Plots and Shapley

Friedman's pioneering concept of partial dependency plots [20] serves to visualize the influence of a single independent variable on the dependent variable, elucidating the marginal effects of these independent variables [26]. Shapley [27] quantifies and visually represents the evolving contributions of individual features within the model. Building upon the foundation laid by PDP, this approach further demystifies the "black box" of machine learning [28,29]. The calculation of Shapley values for features is achieved through the following formula:

$$Shapley(X_j) = \sum_{S \subseteq N \setminus \{j\}} \frac{k!(p-k-1)!}{p!}(f(S \bigcup \{j\}) - f(S)) \tag{1}$$

In the formula, $p$ represents the number of features, $N \setminus \{j\}$ is the set of all possible feature combinations except $X$, $j$ is the set of features in $N \setminus \{j\}$, $S$ is the feature set in $N \setminus \{j\}$, $f(S)$ represents the model prediction corresponding to the feature set $S$, and $f(S \cup \{j\})$ is the model prediction with the feature $X$ in the feature set $S$.

The Shapley value obtained from the model refers to the marginal contribution of a particular feature to the model's prediction. Shapley values can explain many useful properties, such as symmetry, dummy, and additivity, among others [27]. Symmetry means

that if multiple features contribute equally to the model, their Shapley values are equal. Dummy implies that if a feature's marginal contribution to the model is zero, then its Shapley value is also zero. The additivity property requires that the prediction set from a single model equals the predictions from all combined models. In existing research, the methods to calculate Shapley values are primarily based on Monte Carlo approximations and the creation of SHAP [28]. The former can be inaccurate with a small number of iterations, so we opt for the latter, as shown in Equation (2).

$$\hat{y}_i = shap_0 + shap(X_{1i}) + shap(X_{2i}) + \ldots + shap(X_{pi}) \tag{2}$$

In the formula, $\hat{y}_i$ represents the model's prediction result for observation $i$, $shap_0$ is the average prediction value for all observations, and $shap(X_{xi})$ is the SHAP value of the $j$-th feature for the $i$-th observation, indicating the marginal contribution of that feature to the model. Therefore, the SHAP value reflects the degree to which a feature influences the model's predictions and can serve as an importance score for that feature [23].

The use of SHAP can be implemented individually in Python (using the shap library) and R libraries (shapper and fastshap). It can also be used in conjunction with existing machine learning packages, such as scikit-learn, XGBoost, and LightGBM. Current research outcomes predominantly integrate XGBoost and SHAP, while there are fewer achievements in the integration of random forest regression models with SHAP. Additionally, mature integrated algorithm packages such as the "SHAPforxgboost" library in R have not yet been developed. Therefore, it necessitates the development of custom code for combined applications. This process is implemented in the R programming language environment.

### 3.4. Multiscale Geographically Weighted Regression

Ziqi Li [30], based on a generalized weighted model, introduced MGWR. This model can determine the most suitable bandwidth for different variables, enabling regression modeling and producing spatial process models that are closer to reality and more explanatory. The model is formulated as follows:

$$y_i = \sum_{j=1}^{k} \beta_{bwj}(u_i, v_i)x_{ij} + \varepsilon_i \tag{3}$$

where $bwj$ represents the bandwidth used for the regression coefficient of the $j$th variable, $y_i$ denotes the average usage efficiency of charging stations in the $i$th TAZ, $x_{ij}$ denotes the $j$th built environment in the $i$th grid, $(u_i, v_i)$ denotes the center-of-mass coordinates of the $i$th TAZ, $\beta_{bwj}$ denotes the regression coefficient of the $j$th variable in the $i$th grid at bandwidth $bwj$, $k$ denotes the sample size, and $\varepsilon_i$ denotes the random error term. This study was conducted on the GISpro platform, where the domain type was selected as "number of adjacent elements", and the domain selection method was set to "golden search".

## 4. Results

We analyze the spatiotemporal characteristics of user charging behavior based on the charging database. Then, we explore the impact mechanism of the urban built environment on charging behavior.

### 4.1. Extraction and Analysis of Spatiotemporal Charging Characteristics

By extracting the spatiotemporal characteristics of the current usage of charging stations, the analysis results are as follows.

#### 4.1.1. Macroscopic Analysis of User Charging Behavior

(1) Efficiency-time:

From a comprehensive utilization perspective (Figure 3), it is evident that user charging displays three distinct peak periods. These peaks primarily occur just before the morning rush hour (6–7 a.m.), preceding the evening rush hours (15–17 p.m.), and during the late-

night resting period (23–24 p.m.). In essence, users predominantly charge their vehicles during periods of lower road traffic. The temporal utilization patterns of DC fast-charging stations closely mirror the overall usage trends, whereas AC charging station usage remains relatively stable over time. In the case of charging stations equipped with both AC and DC charging facilities, the variations in DC charging usage with time are more pronounced than those of AC facilities. This phenomenon can be attributed to the heightened time sensitivity of users opting for DC fast-charging, as they must consider immediate traffic conditions due to their direct impact on the possibility of queues and congestion during the charging process. Conversely, users who choose AC charging, or slow charging, exhibit diminished sensitivity to immediate environmental factors. Consequently, their charging time selections tend to be more random. This underscores the necessity of incorporating the proportion of fast charging when exploring factors influencing usage efficiency.

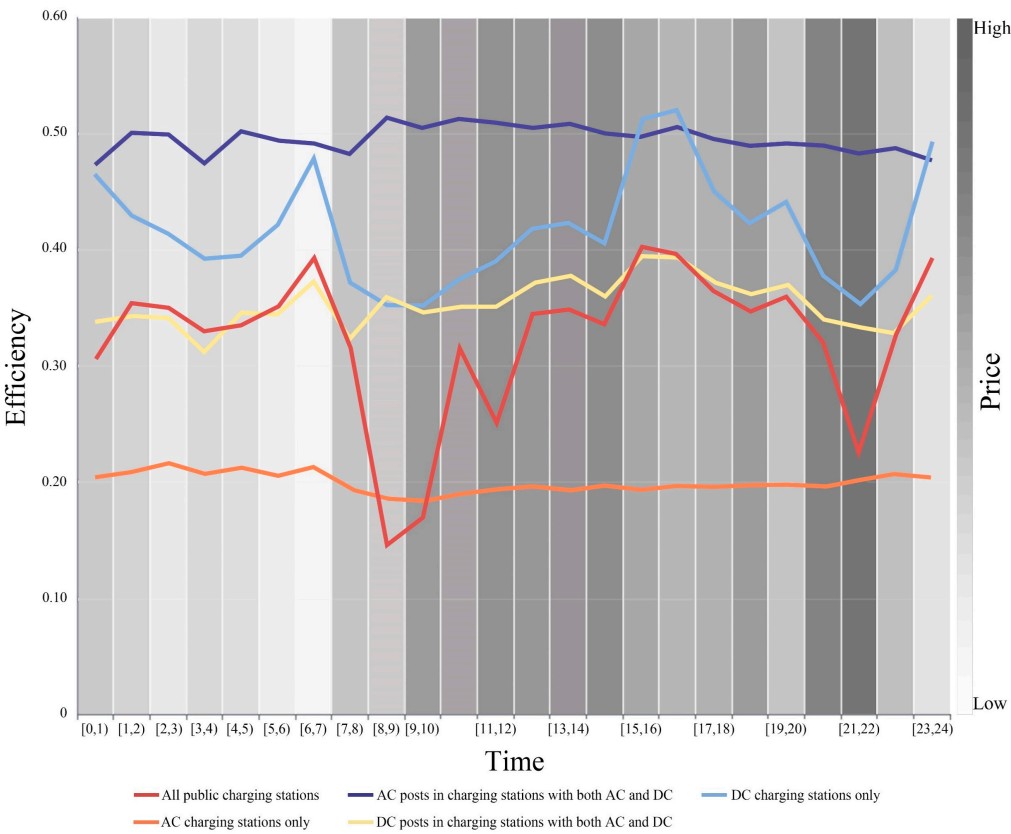

**Figure 3.** Analysis of the relationship between user charging behavior and time/price.

(2)  Efficiency-price-time:

Figure 3 reveals that, from a comprehensive perspective, peak charging behaviors are predominantly observed during time intervals characterized by lower prices throughout the day. Conversely, troughs in charging behavior primarily coincide with time periods of elevated prices. Additionally, during continuous price fluctuations, noticeable surges or declines in the efficiency of charging station utilization become apparent. This underscores users' consistent preference for selecting charging stations with lower prices and highlights the heightened sensitivity of charging behavior to price variations. Notably, between 8 and 10 a.m., even with price increases, the morning rush hour gradually wanes. This implies that users prioritize traffic flow considerations over pricing factors when determining their charging times. Likewise, owing to the significant pricing distinction between fast and slow charging methods, users opting for fast charging exhibit heightened sensitivity in their choice of charging times. In summary, it is evident that, for all public charging stations, the fluctuations in usage efficiency correspond to low- and high-priced periods, respectively,

reflecting the concept of user price elasticity. This underscores the imperative consideration of charging prices as a determining factor in the investigation of usage efficiency.

### 4.1.2. Spatial Characterization of Charging Station Usage Efficiency

Traffic zones were adopted as units for the statistical analysis of charging station usage efficiency, along with a kernel density assessment of the spatial distribution of these stations (Figure 4). Remarkably, the spatial distribution of charging stations exhibits a pronounced clustering pattern, a phenomenon corroborated by the results of spatial autocorrelation analysis (Table 4). This pattern is characterized by a positive spatial correlation. Although Moran's index is relatively modest, the z-score and *p*-value outcomes substantiate the rejection of the null hypothesis, effectively dismissing the possibility of this spatial pattern arising randomly. Consequently, there is a compelling need to investigate the influencing factors and mechanisms underpinning this spatial distribution pattern. Concurrently, observations from Figure 4 reveal a clear association between high-efficiency charging sites and areas with lower kernel densities. Conversely, in regions characterized by higher kernel densities, charging station efficiency levels tend to exhibit uniformity. This phenomenon is attributed to areas with lower kernel densities having a diminished level of charging facility supply. Even when confronted with similar demand, traffic zones with fewer charging stations must accommodate a larger user base, resulting in elevated utilization rates. This underscores the paramount role of charging station supply as a fundamental determinant of user charging behavior.

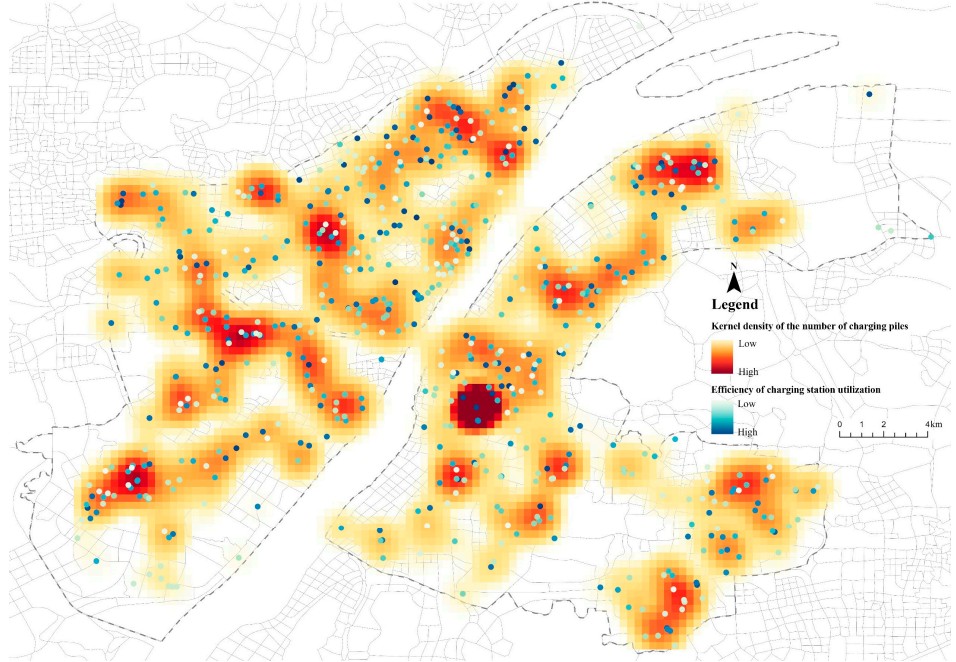

**Figure 4.** Characteristics of spatial distribution of charging stations.

**Table 4.** Global Moran index.

| Moran Index | Expectation Index | Variance | z-Score | *p*-Value |
|---|---|---|---|---|
| 0.0503 | −0.0011 | 0.0002 | 3.4287 | 0.0006 |

### 4.2. Analysis of Influencing Factors of Charging Station Utilization Efficiency

This study first employs multiple machine learning models for regression and selects the model with the best results for further analysis.

### 4.2.1. Model Comparison and Selection

In order to reduce the intensity of sample heteroscedasticity and minimize the fluctuation of variables while adapting to the fluctuation levels of other variables, we performed a logarithmic transformation on the data. Table 5 presents the fitting results of the machine learning regression models. We use $R^2$ and RMSE to describe the model results. $R^2$ indicates the degree of fit of the results, with a higher value signifying a better fit of the model to the data. RMSE is used to measure the average difference between predicted and actual values, with a smaller value indicating greater conformity with reality and a more reliable model. The random forest regression model has the highest $R^2$ and the lowest RMSE, indicating that it performs the best. Therefore, we choose the random forest (RF) model to explore nonlinear relationships and combine it with SHAP and PDP for the interpretation of the "black box" internals, which is our final choice.

**Table 5.** Tree-based machine learning model regression results.

| Model Type | GBRT | XGBoost | RFR |
|:---:|:---:|:---:|:---:|
| $R^2$ | 0.100 | 0.173 | 0.840 |
| RMSE | 13.953 | 13.302 | 5.862 |

### 4.2.2. Nonlinear Relationship

Utilizing the random forest regression model as the foundational framework and leveraging the Shapley algorithm in conjunction with partial dependence plots, we conducted an in-depth exploration of the nonlinear relationship between charging station efficiency and its influencing factors. The outcomes are outlined below.

(1)    Relative importance of factors:

Figure 5 portrays the global importance results, which were computed utilizing the SHAP algorithm. These results provide a comprehensive reflection of the collective impact of various factors on the efficiency of charging stations. Each factor is meticulously ranked in descending order of significance. Notably, vehicle usage demand emerges as the most influential factor, accounting for a substantial 40.74% of the overall influence, significantly surpassing all other factors. Following closely is the DC ratio at 19.87%, trailed by subway accessibility and consumer purchasing power level at 17.04% and 12.31%, respectively. The disparities in importance among the remaining eight factors are relatively marginal.

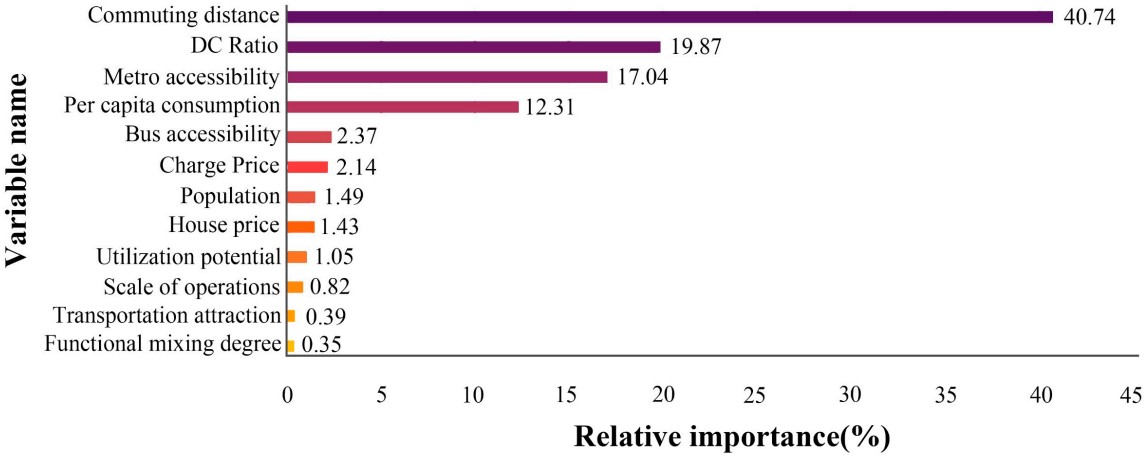

**Figure 5.** Relative importance of each variable based on global impact.

However, it is imperative to acknowledge that, in practice, the influence of factors on efficiency is not a static phenomenon, particularly when considering the collective impact of factors with diverse numerical values. In Figure 6, the horizontal axis represents the computed SHAP values. Higher SHAP values in the positive direction signify a more

substantial positive contribution. Each point on the graph signifies a data point extracted from the database, and their vertical stacking illustrates density, color-coded based on their values. The lower color strip delineates the values of each point, with higher feature values depicted in yellow and lower values in purple. For instance, the red-purple point associated with the charging price factor indicates that lower charging prices (as indicated in the lower color strip) exert a considerable negative impact on efficiency, corresponding to smaller negative values on the *x*-axis.

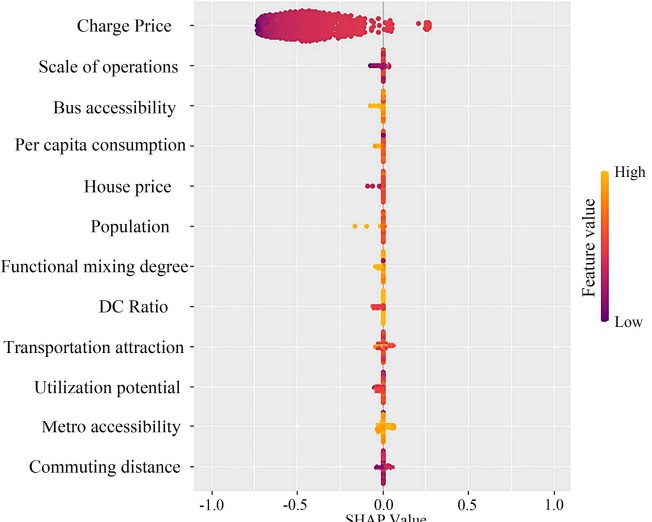

**Figure 6.** SHAP of independent variables based on random forest regression models.

More specifically, Figure 7 elucidates the intricate interplay of factor value fluctuations and corresponding shifts in their importance, underpinned by the SHAP algorithm. Notably, factors such as vehicle usage demand, operational scale, charging price, and housing price exhibit a pattern where, as their values increase, the magnitude of their negative impact on charging station efficiency progressively diminishes, adhering to a logarithmic linear relationship with a base exceeding 1. Conversely, factors such as subway accessibility, potential usage intensity, functional diversity, average consumption level, and traffic attractiveness demonstrate an escalating negative impact as their values increase, forming an inverse proportional relationship with efficiency. Factors like bus accessibility, fast charging ratio, and usage potential exhibit a nuanced pattern where, as their values increase, the importance of these factors initially weakens, subsequently intensifying, predominantly resulting in negative impacts. The relationship between the DC ratio factor and usage potential approximates a sigmoid function, whereas the interaction between usage potential and bus accessibility forms a distinctive reverse U-shaped relationship.

(2)  The relationship between various factors and charging station usage efficiency:

Figure 8 provides a comprehensive illustration of the intricate relationship between independent variables and the dependent variable, shedding light on the multifaceted dynamics influencing charging station efficiency. Vehicle usage demand, recognized as the pivotal variable for efficiency, unveils two distinct inflection points on the graph at 7.94 and 8.81, corresponding to real-world vehicle usage demands of 2.8 km and 6.7 km, respectively. Below 2.8 km, efficiency exhibits gradual growth as vehicle usage demand increases. Within the range of 2.8 to 6.7 km, efficiency experiences a sharp ascent with increasing distance, reaching a peak before displaying fluctuations. Beyond 6.7 km, variations in vehicle usage demand no longer yield significant changes in efficiency.

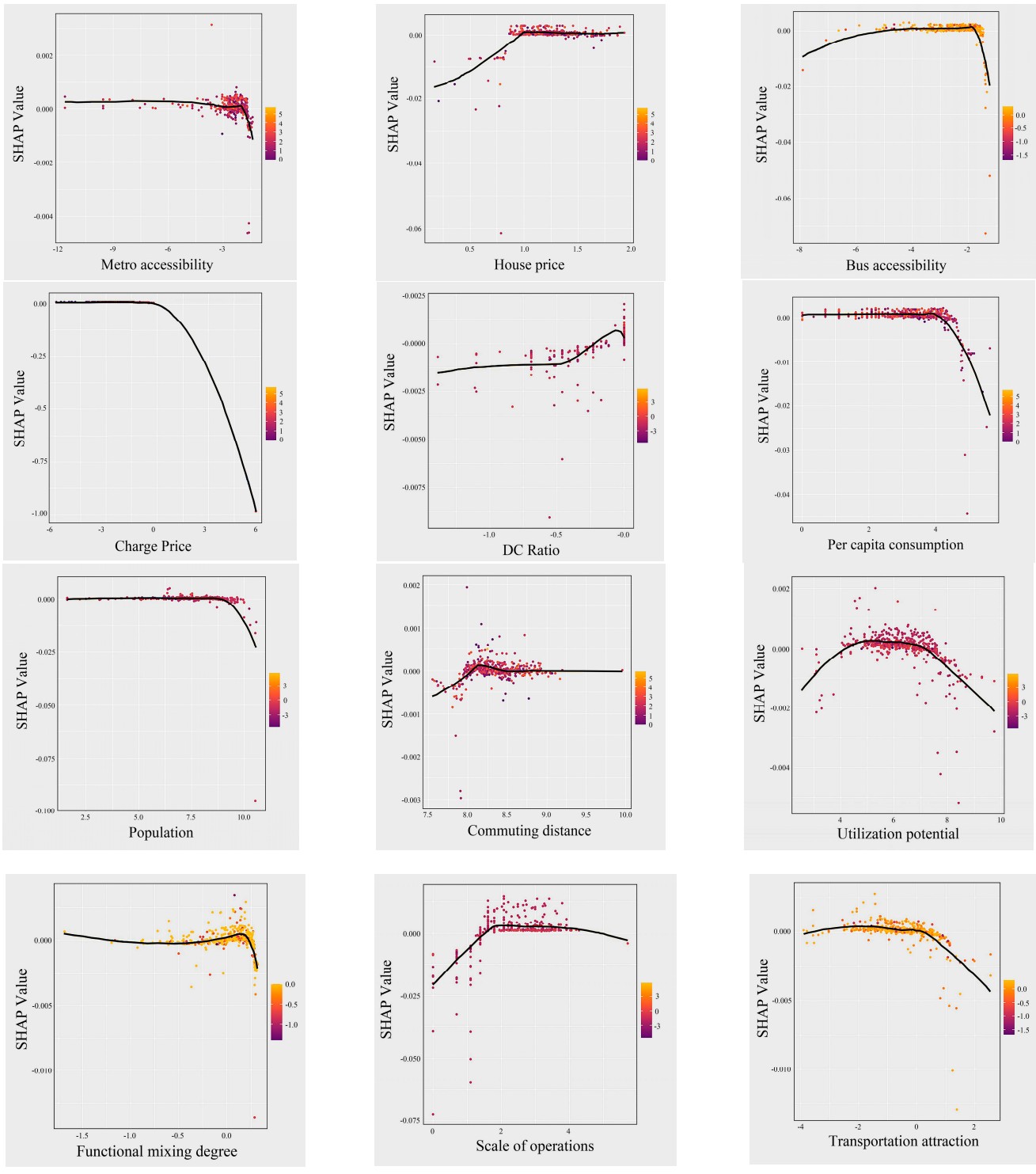

**Figure 7.** SHAP plots for various factors.

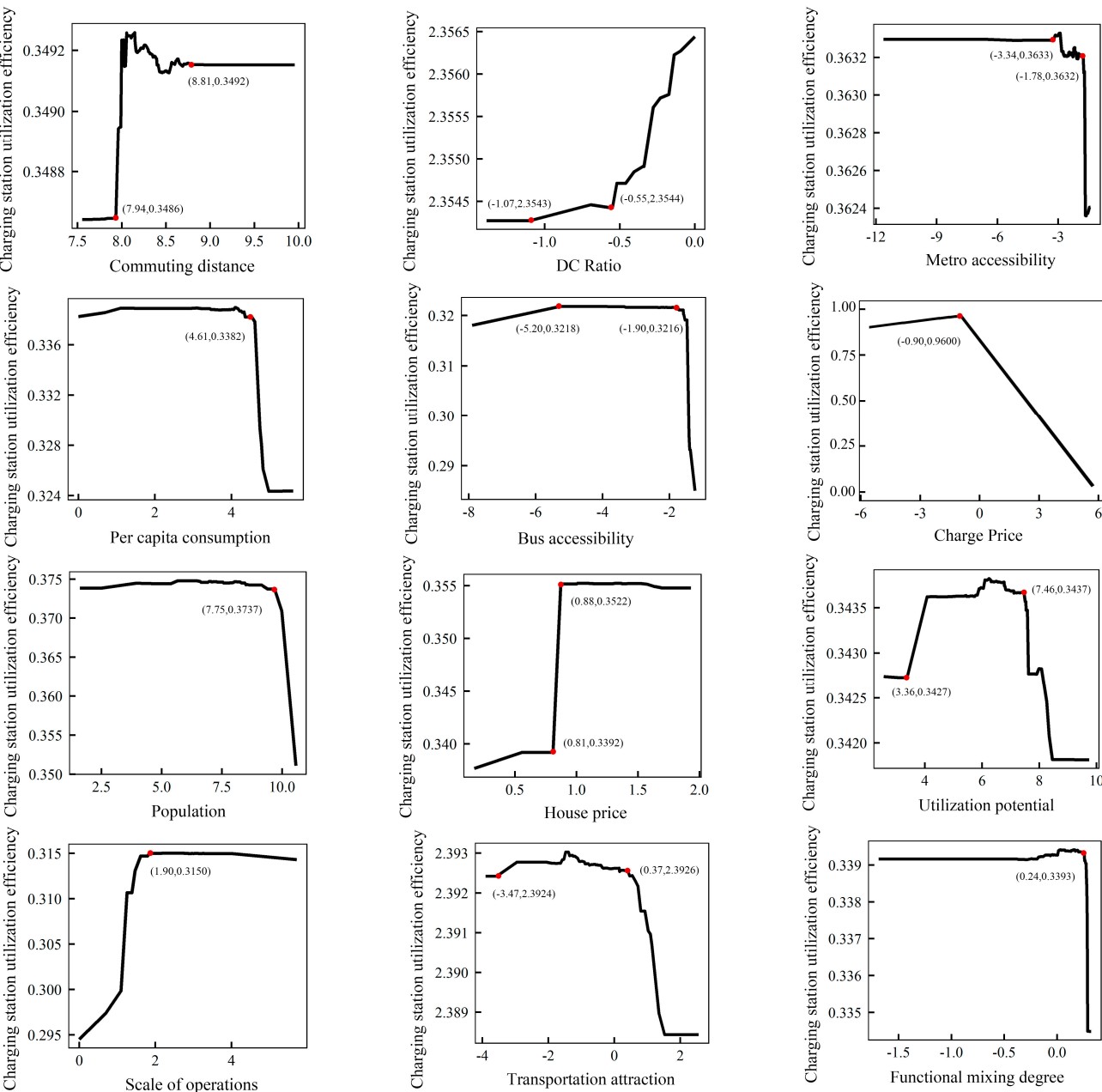

**Figure 8.** Partial dependence plot based on the random forest regression model.

Similarly, DC ratio, the second most influential factor, manifests two critical inflection points at −1.07 and −0.55, corresponding to actual ratios of 89.21% and 56.67%, respectively. Efficiency remains constant when the ratio exceeds 89.21%, while within the 56.67% to 89.21% range, efficiency progressively improves as the ratio diminishes. This observation aligns with the coefficient changes observed in the MGWR analysis. Below 56.67%, as the ratio continues to decrease, efficiency experiences a swift, pronounced upsurge.

In the context of metro accessibility, two inflection points correspond to real values of 0.07 and 0.20. When accessibility falls below 0.07, efficiency remains relatively stable. Within the range of 0.07 to 0.20, efficiency displays reduced fluctuations. Once accessibility surpasses 0.20, efficiency experiences a sharp decline. The inflection point for the average consumption factor is noted at 4.61, corresponding to a real value of 100 yuan per person. Prior to surpassing this threshold, efficiency remains relatively steady. However, once this point is exceeded, efficiency follows a nearly linear descent, stabilizing at a lower level.

Turning our attention to the remaining eight influencing factors with relatively minor variations in importance, such as bus accessibility, charging price, potential usage intensity, traffic attractiveness, and functional diversity, a consistent pattern emerges. Initially, efficiency remains relatively stable, but once a critical threshold is crossed, it precipitously plummets to its lowest point, maintaining that level. In contrast, the factors of housing price and operational scale display a gradual increase followed by a sudden peak, leading to sustained high efficiency levels. Notably, the usage potential factor augments the previous pattern with a subsequent stage of stepwise decline, ultimately stabilizing at the lowest level.

## 5. Discussion

### 5.1. Nonlinear Effects of Built Environment on Charging Station Use

Commuting distance, as the most important variable affecting the efficiency of charging station usage, shows that the further the commuting distance, the greater the efficiency of the charging station. This is because longer commuting distances make nonmotorized modes such as walking and biking less feasible [31], thus increasing the demand for motor vehicles and consequently the frequency of charging. However, when the distance exceeds a certain threshold, destinations that are too far do not promote charging behavior, which differs from existing research conclusions [31]. This is because, on one hand, long commuting distances provide a wider range of charging stations to choose from [32], reducing the likelihood of using charging stations within the residential area. On the other hand, considering economic and time costs, very long distances increase the likelihood of using mixed-mode commuting [33], likely owing to the proliferation of charging options along longer routes and the adoption of multimodal transportation methods, which in turn reduces the proportion of electric vehicle usage and weakens the impact of commuting on charging station usage. This nuanced understanding both challenges and extends previous research, highlighting the complexity of urban mobility patterns.

Charging speed, as the second most important variable affecting efficiency, shows that an increased proportion of fast charging enhances the efficiency of charging stations, indicating that more convenient charging methods are more attractive to users. However, when the proportion of fast charging at a station is too high and the supply of cheaper slow charging is insufficient, it can adversely affect the overall efficiency of the charging station, which differs from existing studies [14,34]. This is because, for stations offering both fast and slow charging, a higher proportion of fast charging equates to higher efficiency. In contrast, stations offering only fast charging are less efficient due to higher prices and lack of cheaper alternatives, and their efficiency is very sensitive to this single mode of charging.

Metro accessibility, the third critical factor, exhibits an inverse relationship with EV charging demand. Regions with improved subway access observe a decrease in EV usage [35]. This indicates a competitive relationship between public transit availability and EV adoption. These insights are crucial for urban planners and policymakers, especially in integrating EV infrastructure with public transportation networks to promote sustainable urban mobility [36].

Regions with higher consumption levels show lower charging station efficiency [37], possibly due to more transportation options and prevalent traffic congestion [38,39]. Additionally, the study identifies clear price elasticity in charging behavior; when charging prices surpass a certain point, station usage sharply decreases. This highlights the importance of pricing strategies in influencing EV adoption and charging station usage.

The correlation between higher housing prices and increased charging station efficiency, in line with studies emphasizing the prevalence of public charging infrastructure in affluent areas [38,39], suggests a socioeconomic aspect to EV adoption. Higher income levels frequently correlate with increased environmental consciousness and the financial ability to own EVs, potentially explaining this trend. This relationship merits further investigation into how economic status affects sustainable transportation choices. The findings suggest that users prefer charging stations in more desirable locations with higher

usage potential [40]. However, excessive traffic resulting from crossing a usage threshold negatively impacts charging station efficiency. This highlights the need for strategic urban planning that balances accessibility and traffic management for optimal charging station placement. The study shows a positive correlation between the operational scale of charging stations and their efficiency, indicating that stations built for substantial demand can optimize usage [35]. Although increased traffic flow indicates a high potential user base, it paradoxically deters charging station use due to the preference for quick and convenient charging options [40,41]. Regions with varied business formats attract higher charging demand by offering accessibility to multiple destinations [35]. However, this functional diversity's downside is potential traffic congestion, negatively affecting charging station utilization. This indicates the need to integrate commercial development with strategic transportation planning for efficient EV charging infrastructure use.

*5.2. Discussion of MGWR-Based Spatial Optimization Strategies*

This study employs a rigorous methodological approach, including a detailed collinearity diagnosis and an R-squared value of 0.665, demonstrating the robustness of the multi-scale geographically weighted regression (MGWR) model. The model's exceptional ability to manage spatial heterogeneity and interactions between variables renders it particularly adept at analyzing the complex dynamics that affect the efficiency of electric vehicle (EV) charging stations. It effectively captures the intricate spatial variations and collinear relationships among diverse urban factors, offering a comprehensive understanding of the determinants of charging station efficiency.

Utilizing MGWR, the study explores spatial variations in key factors that impact the efficiency of charging stations. It uncovers significant spatial differences in vehicle usage demands, particularly in areas with distinct functions, such as industrial or residential zones [42,43]. These differences directly influence the efficiency of EV charging stations. For example, in Optics Valley (Zone A), it is recommended that the deployment of charging stations be aligned with workforce commuting patterns to enhance efficiency. This strategy aligns with wider urban development goals, promoting sustainable and effective infrastructure utilization.

In regions like Zone A, marked by industrial activities, and Zone B, primarily residential, the increased demand for commuting significantly affects the use of charging stations. This effect is further heightened by the limited availability of alternative transportation options, such as subways and buses, as shown in Figure 9c,d [42,43]. The relationship between commuting patterns and transportation choices highlights the necessity for integrated urban planning that encompasses various mobility solutions.

Charging station placement optimization strategies, especially in critical areas such as Optics Valley, are centered on harmonizing with user commuting behaviors [44,45]. These strategies not only aim to improve charging facilities in these areas but also consider environmental and social equity aspects. The goal is to achieve a balance between charging convenience, ecological impact minimization, and accessibility for all community members. Additionally, the current placement of charging stations, closely linked with existing parking facility developments, presents an opportunity for strategic urban resource optimization. In Area A (Figure 10a,b), the availability of unused parking spaces near efficient charging stations indicates room for expansion. However, this expansion should be thoughtfully planned, considering local demographics, commuting destinations, and the distribution of private chargers [46]. This method ensures that new installations align with actual demand and contribute to the overall effectiveness of the urban transportation system. Similarly, in Area B (Hanyang), which is primarily residential, charging mainly occurs before work and after returning home, considering commuting. Before work, to reduce time costs, there should be an increase in the proportion of fast charging stations. After returning home, when there is more time, slower charging at a relatively lower price is preferable. However, considering construction costs, indiscriminate incremental construction is not feasible. Instead, promoting the sharing of private charging piles is

advisable. Idle private charging piles can be integrated into the market and managed uniformly, allocating charging demand reasonably according to relevant requirements. This approach can also bring additional economic benefits to the owners of charging piles, achieving a win-win situation.

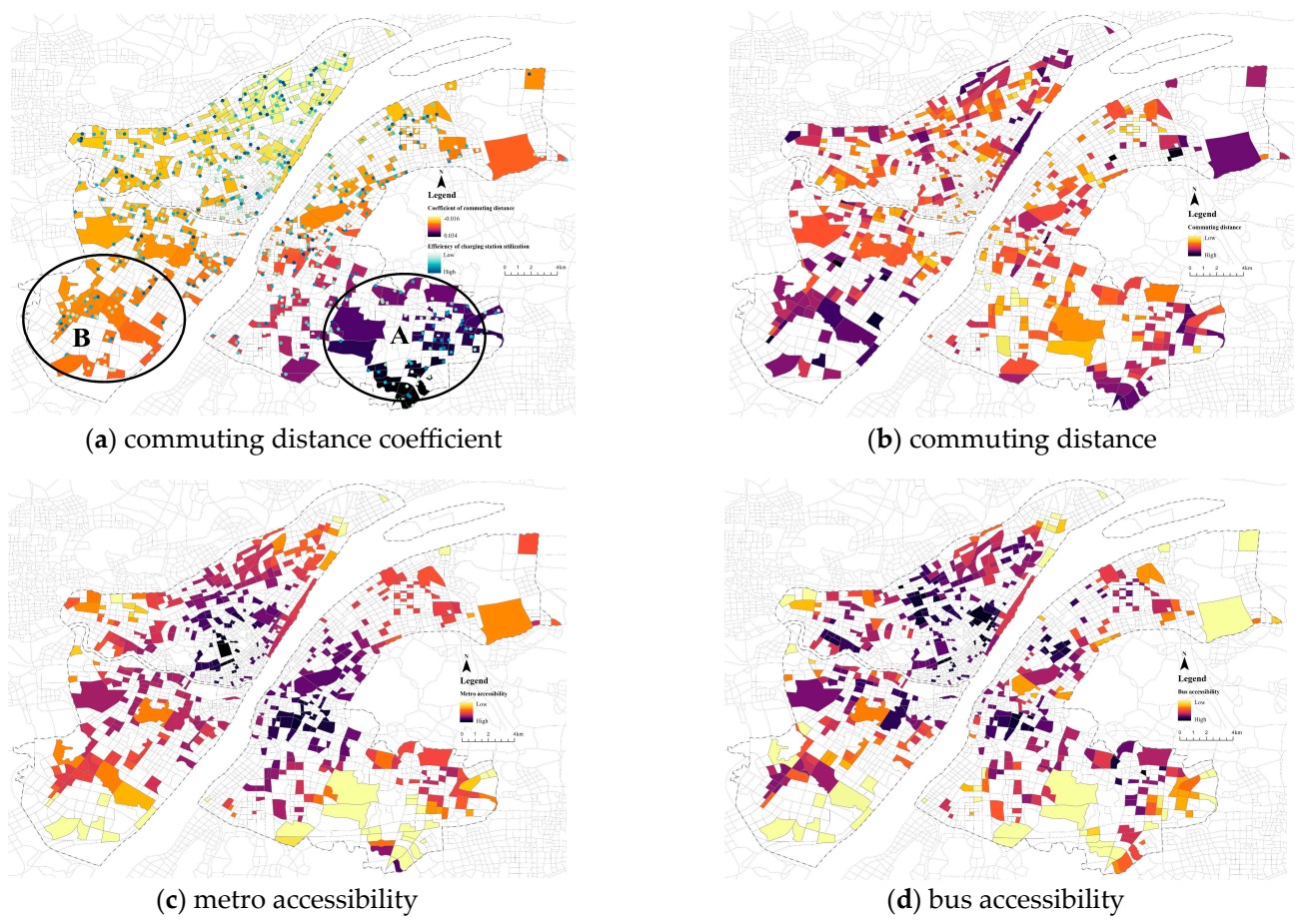

(**a**) commuting distance coefficient

(**b**) commuting distance

(**c**) metro accessibility

(**d**) bus accessibility

**Figure 9.** Spatial patterns of vehicle usage demand coefficients and the distribution of related factor values.

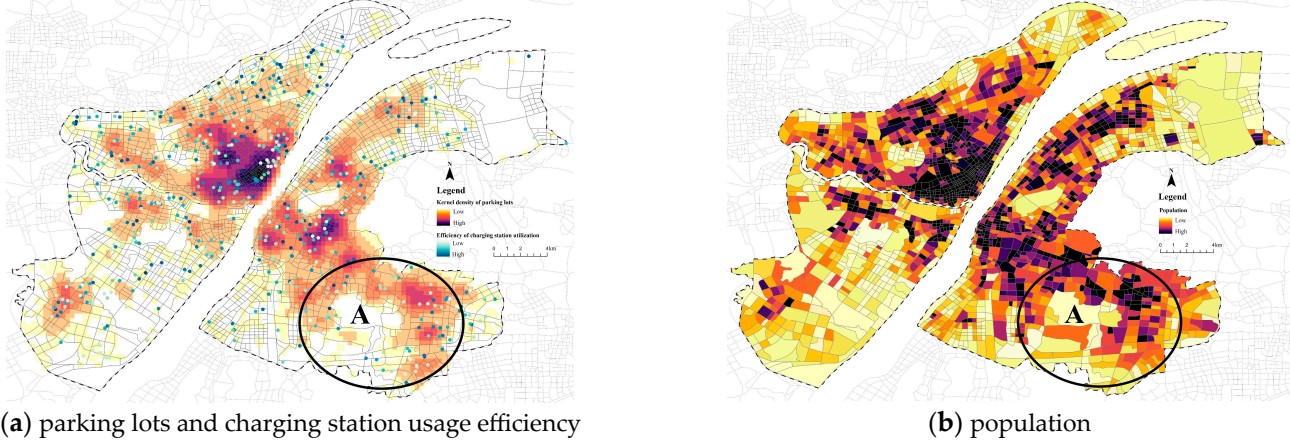

(**a**) parking lots and charging station usage efficiency

(**b**) population

**Figure 10.** Spatial distribution of parking lots and potential usage intensity.

## 6. Conclusions

This study represents a significant advancement in understanding the spatiotemporal characteristics of EV charging station usage. By analyzing millions of charging records, it overcomes the limitations of smaller datasets used in previous studies. Compared with existing research, this study presents many supplementary conclusions. It delves deeper into the exploration of marginal effects between influencing factors and the usage of charging stations, providing scientific support for investment and resource regulation in the construction of charging stations.

Employing a random forest regression model, the study elucidates the nonlinear relationships between various urban factors and charging station efficiency. The SHAP algorithm and partial dependence plots offer detailed explanations of these relationships, highlighting the importance of considering multiple factors in the design and placement of charging stations. The study's use of advanced MGWR modeling further refines the spatial analysis, allowing for targeted optimization strategies based on vehicle usage demand and other critical variables.

The findings underscore the complex interplay between the built environment and EV charging station efficiency, challenging traditional linear perspectives. This research contributes to the broader field of urban sustainability by providing empirical evidence and methodological approaches that can inform the construction and optimization of EV charging infrastructure. The results offer a scientific basis for investment decisions and resource allocation in this rapidly evolving sector.

However, the study is not without limitations. The data employed exhibit temporal incongruities, and the three-month dataset used primarily represents summer months, potentially influencing the findings. Future research will expand the data collection period and consider seasonal variations in charging behavior. Additionally, future studies will aim to incorporate micro-level analyses, such as agent-based simulations, to gain deeper insights into user charging preferences and behavior patterns.

In conclusion, this research makes a substantial contribution to the field of smart city technologies and urban sustainability, offering practical insights and strategies for the efficient deployment of EV charging stations in urban areas. It paves the way for future studies that will further refine our understanding of the dynamic relationship between urban environments and sustainable transportation infrastructure.

**Author Contributions:** Conceptualization, Chenxi Liu and Lingbo Liu; methods, Chenxi Liu and Lingbo Liu; software, Chenxi Liu and Hao Wu; validation, Chenxi Liu, Zhenghong Peng and Hao Wu; formal analysis, Hao Wu; resources, Lingbo Liu; data organization, Chenxi Liu; writing—original draft preparation, Chenxi Liu; writing—review and editing, Zhenghong Peng; visualization, Chenxi Liu and Hao Wu; supervision, Zhenghong Peng and Hao Wu; funding acquisition, Zhenghong Peng and Hao Wu. All authors have read and agreed to the published version of the manuscript.

**Funding:** This research was funded by National Natural Science Foundation of China, grant numbers 52078390 (Hao Wu) and 51978535 (Zhenghong Peng).

**Data Availability Statement:** Electric vehicle charging data presented in this study are openly available in "Power Up" App; Urban statistical information are openly available in China statistical yearbook, house price platform and mass appraisal APP; POI, travel records, and city road data presented in this study are openly available in Amap platform; Mobile signal data presented in this paper are not readily available because of limitations in China's privacy protection policies and authorization of third-party data sources.

**Conflicts of Interest:** The authors declare no conflicts of interest.

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
