# Peer review of "Analysis of Spatiotemporal Characteristics and Influencing Factors of Electric Vehicle Charging Based on Multisource Data"

_ijgi, doi:10.3390/ijgi13020037_

Round 1

Reviewer 1 Report

Comments and Suggestions for Authors

The paper analyses spatiotemporal characteristics and influencing factors of electric vehicle charging based on multisource data. The paper's content is well expressed; however, the presentation of the paper should be further improved. Some comments are given as follows. 

 1. The abstract is too long, and it is suggested to revise it in shortened and solid form. 

2. Table 1 should be put on the same page, do not put it over two pages. 

3. The methodology should be written more understandably. Only having two equations is not enough, how is the method implemented to obtain the results?  This should be explained clearly. 

4. See Table 3, the declaration of indicators should be written appropriately, this is to make a difference to the sentences in the description column. 

5. See page 7, line 243, what are "ntree=100, mty=3, method = "rf,""? These are the commands used in the R language, please mention these commands more understandably. 

6. See page 7, notations and symbols should be written in italic form. Please check. 

7. The caption for Figure 3 has a typo, please check. 

8. See Table 5, what are the terms "R and RMSE"? Please define. 

9. Where is Figure 6? We only see the caption of the figure. Please check.

Author Response

Dear Reviewer,

I hope this email finds you well. I am writing to extend my heartfelt gratitude for the time and effort you dedicated to reviewing my manuscript. Your insightful comments and constructive feedback have been immensely beneficial to the refinement of my work.

I have carefully considered each of your suggestions and have made corresponding revisions to the manuscript. To provide a clear and detailed response, I have compiled all the changes and my responses to your comments in an attached document. Please find the attachment for your perusal. 

Once again, thank you very much for your valuable contribution to my work. Your expertise and thoughtful guidance are greatly appreciated.

Warm regards,

All authors

Reviewer 2 Report

Comments and Suggestions for Authors

This is an interesting paper but it needs more work. The authors set out to examine spatiotemporal characteristics of  EV user  charging behavior and the impact of the spatial environment on the observed behaviors. As the authors note, a better understanding of these matters could help in optimizing the configuration and layout of charging stations. The paper uses state of the art methods and reasonable data. However, as the paper is currently written, the specifics of their research are unclear.

1-      The authors should describe the case study area in more detail so that international readers have a better understanding of the context and can better infer whether this case is relevant to their own situations.  In particular, it would be helpful to know what percent of the population has an EV, what the characteristics of EV owners are compared to the population in general, what it costs to charge an EV and how this relates to EV owner income, etc.

2-      The  authors need to edit their text to present a clearer picture of their database.  On p. 3 they say they have data from a million stations but on p. 4 they say they have data from 130,000 stations and after focusing on public charging stations  and otherwise eliminating a few problematic sites they say they have 858 stations. They need to present a consistent, correct, description.

3-      The authors say:

The dataset includes user call records, which capture user calling behavior and information about base stations. To begin, we categorized the travel data,  extracted user age information, and conducted a summary analysis of the demographic 1age distribution within specific spatial units. Recognizing the high density of base stations  in the central urban area, we filtered out data points with travel distances less than 100 meters. This process resulted in a dataset of 3,037,616 valid records and encompassed 1,515 spatial units with valid data. Subsequently, we determined the spatial locations of user call stations and the corresponding time periods of user calls. Building upon this  information, we demarcated spatial units representing user residences and workplaces, enabling us to summarize the population density within these temporal and spatial segments. Lastly, we computed the distances between these home and workplace spatial units, ultimately deriving the average commuting distance for the population within these  designated spatial units. Detailed sample data is presented in Table 2.

I can see how the authors could determine travel distances  by time of day from cell phone data, but would like to hear more about how they determined whether a call was from a residence or workplace, (especially since many locations are mixed use), how that enabled the  authors to “summarize” (do they mean estimate?) the population density in each location, and how they determined the mode of travel being used. How do they determine whether a cell phone user is also an EV user?   I also would like more information on where they got user age and other demographic information.

4-      The authors say:

Taking into full account the elements in the built environment that pertain to charging station utilization and drawing from extant literature [11–15], this paper formulates 12 indicators across three dimensions: charging station self-configuration, surrounding  traffic conditions, and regional demand levels, with the objective of scrutinizing the impact of the built environment on charging station efficiency. Table 3 presents descriptions  and calculation methodologies for all these indicators. Unfortunately, Table 3 is insufficient to give the reader a clear understanding of why the authors hypothesize that each variable belongs in the model, nor does Table 3 or the text  adequately explain what the variables signify, how they are measured, or what impact the authors expect each variable would have on EV charging behavior.   This needs much more explanation. In particular:

·       Why use average price as a measure rather than e.g. using a range measure (differential of  price for slow charging price and price for fast charging)?

·       Why is housing price expected to affect user charging behavior?  Is it intended to be a proxy for income, or is there some other justification?

·       What do the authors mean by integrability?

·       What do they mean by utilization potential and selectivity?

·       Where did they get data on restaurant consumption, how did they measure consumption, and what do they hypothesize this means wrt  EV charging? 

·       It's not clear what the authors mean by charging station efficiency in this context.

5-      The results section is more descriptive than analytical – it says what the results were but does not provide any behavioral insights. It also raises some questions about how the analysis was actually done, e.g., it appears that the authors have information about price differences by location and time of day, but their description of their data and analysis methods offers no insight into how they actually did this analysis. An example of the authors being descriptive but failing to provide a meaningful analysis is as follows:

Turning our attention to the remaining eight influencing factors with relatively minor

variations in importance, such as bus accessibility, charging price, population, traffic attractiveness, and functional diversity, a consistent pattern emerges. Initially, efficiency remains relatively stable, but once a critical threshold is crossed, it precipitously plummets to its lowest point, maintaining that level. In contrast, the factors of housing price and operational scale display a gradual increase followed by a sudden peak, leading to sustained high efficiency levels. Notably, the usage potential factor augments the previous pattern with a subsequent stage of stepwise decline, ultimately stabilizing at the lowest level.

6-      The discussion could be improved by focusing on (and sharpening) the research question. What behavioral insights can the authors glean from their data?   For example, are Wuhan EV owners more likely to use a fast charge just before going to work or just before going home, and more likely to use a slow charge at home locations and after work? 

Comments on the Quality of English Language

The paper is readable as is.

Author Response

(The authors gave the same response as above.)

Reviewer 3 Report

Comments and Suggestions for Authors

The paper is in the area of electric vehicles and addresses a theme that can be considered "hot" now-a-days.

First, I think that the abstract should be reduced as it is far outside the recommendations of the journal and more emphasis should be given to the results and policy recommendations.

The introduction is easy to read and highlights the need for the current study.

Please, if possible, add here some of the research questions the study tries to address.

Please provide a source/reference for Figure 1.

Please choose a better title for section 3.3., particularly please replace "explainable".

Please add some text between each section/sub-section and the following one.

Please increase the readability of the figures in the paper.

In my opinion, the paper is too oriented to the considered case study. Even though the methodology is of interest, I think that a more generalized approach should be given when presenting the methodology in order to ensure that it can be replicated in other similar situations. Nevertheless, more information should be provided in the methodology section as in the current form it cannot be reproduced, as the core part of the process is  only presented briefly.

Some validation of the proposed approach would be of interest.

Also, a comparative discussion with other approaches from the field would be of interest.

Nevertheless, a comparison in terms of results with other studies would be more than appreciated.

Minor:

- section 4 should be "results" instead of "result"

- please check grammar, spelling and typos - e.g. add a dot in row 267.

- please use the citations style recommended by the journal - please place the reference near the name of the authors - e.g. row 266  instead of "Fotheringham and others" please use "Fotheringham et al. [30]"

Author Response

(The authors gave the same response as above.)

Round 2

Reviewer 3 Report

Comments and Suggestions for Authors

Thank you for the revised version of the paper. I have no further comments.